Unravelling the potential of nitric acid as a surface modifier for improving the hemocompatibility of metallocene polyethylene for blood contacting devices

Vellayappan Muthu Vignesh
Jaganathan Saravana Kumar jaganathaniitkgp@gmail.com
Muhamad Ida Idayu
IJN-UTM Cardiovascular Engineering Centre, Faculty of Biosciences and Medical Engineering, Universiti Teknologi Malaysia , Malaysia
Palomo Jose
Electronic publication date: 2016 Jan 19
Publication date: 2016
Volume: 4
Electronic Location ID: e1388
Received 2015 Jul 13; Accepted 2015 Oct 16
Copyright: ©2016 Vellayappan et al.
Copyright year: 2016
Copyright holder: Vellayappan et al.
License: This is an open access article distributed under the terms of the Creative Commons Attribution License, which permits unrestricted use, distribution, reproduction and adaptation in any medium and for any purpose provided that it is properly attributed. For attribution, the original author(s), title, publication source (PeerJ) and either DOI or URL of the article must be cited.
License URL: https://creativecommons.org/licenses/by/4.0/

Keywords: Hemocompatibility, Surface modification, Nitric acid, Blood contacting device

Funding: GUP Universiti Teknologi Malaysia Q.J130000.2509.10H13 This work was supported partly by GUP Universiti Teknologi Malaysia with the Grant Vot No:Q.J130000.2509.10H13. The funders had no role in study design, data collection and analysis, decision to publish, or preparation of the manuscript.

==============================
Design of blood compatible surfaces is obligatory to minimize platelet surface interactions and improve the thromboresistance of foreign surfaces when they are utilized as biomaterials particularly for blood contacting devices. Pure metallocene polyethylene (mPE) and nitric acid (HNO3) treated mPE antithrombogenicity and hydrophilicity were investigated. The contact angle of the mPE treated with HNO3 decreased. Surface of mPE and HNO3 treated mPE investigated with FTIR revealed no major changes in its functional groups. 3D Hirox digital microscopy, SEM and AFM images show increased porosity and surface roughness. Blood coagulation assays prothrombin time (PT) and activated partial thromboplastin time (APTT) were delayed significantly (P < 0.05) for HNO3 treated mPE. Hemolysis assay and platelet adhesion of the treated surface resulted in the lysis of red blood cells and platelet adherence, respectively indicating improved hemocompatibility of HNO3 treated mPE. To determine that HNO3 does not deteriorate elastic modulus of mPE, the elastic modulus of mPE and HNO3 treated mPE was compared and the result shows no significant difference. Hence, the overall observation suggests that the novel HNO3 treated mPE may hold great promises to be exploited for blood contacting devices like grafts, catheters, and etc.

Introduction

The surface modification of biomaterials is a process of modifying its surface properties by changing its inherent physical, chemical or biological properties to possess desirable characteristics (John et al., 2015). Generally, the surface modification of biomaterials can be done via different techniques for the biocompatibility enhancement, which is the cornerstone property required whilst selecting a blood contacting device (Jaganathan et al., 2014a; Vellayappan et al., 2015a; Li et al., 2010). There is a wide range of blood contacting devices available nowadays like grafts, catheters, hemodialysis, bypass/extracorporeal membrane oxygenation, and ventricular assist devices (VADs). Even though there is a widespread need for blood contacting devices, the formation of blood coagulation as well as commencement of thrombotic events whilst the biomaterial comes in contact with the blood, remains as a daunting challenge for researchers to decipher (Vellayappan et al., 2015a; Vellayappan et al., 2015b). A recent statistic shows that 65–88% of aortic repair procedures performed in the US are being replaced with endovascular grafts and the thrombus formation in aortic side branches often leads to ischemia (Thompson, 2013). Another clinical study dictates that thrombus formation on the catheter surface in 50% of patients undergoing diagnostic angiography (Formanek & Frech, 1970). Moreover, thrombosis is found to be the precipitating event in 30–40% of central venous catheter malfunctions (Vascular Access, 2006). Thus, prevention of thrombotic deposition and occlusion, triggered by the activation of the coagulation cascade and platelets, is a mandatory property which the implanted blood contacting devices should possess before it is recommended for clinical trials.

The advent of latest technology has paved the way for the discovery of novel polymers like metallocene which is a new class of polyolefins with superior performance characteristics like improved toughness, sealability, clarity, and elasticity. Metallocene is made up of two cyclopentadienyl anions (Cp,) which are attached to a metal center (M) with an oxidation state II, hence resulting in a general formula M(C5 H5)2 (Kealy & Pauson, 1951). The metallocene polyethylene (mPE) is one of the versatile polymers. The mPE has wide spectrum of applications in disposable bags, storage bottles, blood bags, and syringe tubes. Albeit mPE has an excellent permeability to oxygen and functions as an effective barricade towards ammonia and water, yet mPE poor blood compatibility hampers it from being used for blood contacting devices (Mohandas et al., 2013). Thus, different works were done for enhancing the blood compatibility of mPE recently to promote it for various biomedical applications.

In our group, we are exploring several modification techniques to improve the blood compatibility of mPE. Recently, Mohandas et al. (2013) utilized microwave radiation for surface modification of the mPE to improve its blood compatibility. Furthermore, the effect of hydrochloric (HCl) acid treatment on the metallocene polyethylene mPE depicted an enhanced blood compatibility of mPE compared to the untreated mPE sample (Jaganathan et al., 2014b).

Since the HCl etching effect on mPE yielded good results, it further motivated us to find other available substitutes which are cost effective and easily available for improving the blood compatibility of mPE. Thus, being a very strong acid and oxidizing agent, we have selected HNO3 for improving the blood compatibility of mPE. In work done utilizing HNO3 by Moreno-Castilla et al. (1995) dictates that the HN03 treatment affects the surface area of activated carbons and their porosity the most compared to the other treatments like hydrogen peroxide, and ammonium peroxydisulfate treatments. Moreover, Dong et al. (2013) demonstrated the HNO3 oxidation treatment on CNT modifies the CNTs physical and chemical properties resulting in improved CNTs biocompatibility. For the first time the effect of HNO3 treatment on mPE is documented in this work. Furthermore, the present study is done to ascertain the modifications induced in mPE and its impact on the blood compatibility of the HNO3 treated mPE samples.

Materials and Methods

Ethics statement

The blood coagulation assays were carried out in India and the characterization tests were done in Malaysia. Prior to blood procurement, the written consent form was given to the healthy volunteers. They read the benefits and risks of participation before expressing his/her willingness by signing the form. All protocols of blood procurement and consent procedure were approved by the Pacheri Sri Nallathangal Amman (PSNA) College of Engineering and Technology Ethical Committee of Dindigul with an approved IRB number: H30114. Later, the blood was extracted via venipuncture from aspirin-free healthy adult human donor and it is prevented from coagulation with trisodium citrate at a volumetric ratio of 9:1. Newly prepared platelet rich plasma (PRP) was acquired from the Dindigul Blood Bank, Dindigul, India.

Sample preparation and acid treatment

Initially, two mPE films of dimensions 10 cm × 10 cm were cut into two samples with a size of 1 cm × 1 cm. Then the samples were washed with 70% ethanol and distilled water prior subjecting it to HNO3 treatment. Then, 8–10 ml of concentrated HNO3 with molarity of 15.9 M was poured into petri dish which contains the square shaped sheets of mPE. The acid and sample containing dishes were later placed on the rocking shaker which moves at a constant speed. Moreover, in this work mPE sample were subjected to HNO3 exposure for the different time durations. From that, the optimized timings were selected by observing surface changes with an optical microscope at 40×. The samples subjected for a lesser duration didn’t confer notable surface modifications when compared with control, however, during 30 min of exposure significant change in the surface of mPE was observed. While subjecting samples for a prolonged period, changes noted, were not significant compared to 60 min treated sample. Thus, for characterization studies, 30 min and 60 min treated samples alone were considered. Once the acid treatment was done, the samples were washed in distilled water and dried at room temperature overnight before performing any characterization tests. Whilst preparing the samples for blood compatibility tests, samples were kept in a beaker with physiological saline and then in a rotary shaker overnight at 37 °C to remove the acid present on the surface of the polymer.

The scheme of the experiments performed was shown in Fig. 1.

Figure 1 Schematic representation of series of characterization and blood compatibility experiments done.

Characterization of the samples

Contact angle measurement

The hydrophilicity tendency of the polymer was determined using Dynamic Contact Angle Analyzer (FTA200—First Ten Angstroms). Here, a water droplet was placed on the surface of the sample. Water droplet of 1 µL was used and the photographs were taken in the ultra-fast mode within 30 s. The degree of the angle formed is determined using computer interfaced software. The contact angles were recorded and analyzed for the samples which are untreated mPE, 30 min and 60 min HNO3 exposed samples (n = 3).

Attenuated total reflectance fourier transfer infrared spectroscopy (ATR-FTIR)

ATR-FTIR equipment NEXUS- 870 model spectrophotometer was utilized have additional features such as extended beam splitter, two light sources, and middle band MCT detectors with various sampling options. This was used for the purpose of analyzing the chemical compositions or functional groups present within the polymer. There are three samples present in the study, which are untreated mPE, 30 min and 60 min HNO3 treated mPE samples. All these samples were studied using this ATR-FTIR.

3D-Hirox digital microscope

The latest 3D-Hirox digital microscope model (KH-8700) was used to determine the formation of pits and pores of the samples. The 3D-Hirox digital microscopy images are very useful in determining the morphological structure of samples to determine whether the sample has pores or it has an even surface. There are two types of images which are obtained from 3D-Hirox digital microscopy either with or without profilometry line. The surface morphology of 1 cm × 1 cm of mPE and HNO3 treated mPE sample was assessed at an area of approximately 5, 757μ2 at a magnification of 500×. The same as white light confocal profilometry, in-focus and 3D images were obtained using this 3D microscope. Slices of the image were captured at different heights acquired for the surface topography analysis (Pereira et al., 2013). Maximum of three profiling lines is chosen as the profiling value of the each sample. Each point in X, Y and Z axes of the profiling line is measured and their values can be exported in excel sheet to represent the height of the pits in the sample. Data processing was performed using the in-build 3D profilometry software. Images were recorded at standard 1,200–1,600 pixel resolution.

Scanning electron microscope

The surface microstructure of the samples can be critically analyzed in detail by using SEM. The SEM which is utilized to study the polymeric samples was JEOL JSM5800 SEM with OXFORD ISI 300 EDS X-ray Microanalysis System. Untreated mPE, 30 min and 60 min HNO3 treated mPE underwent gold sputtering and then been studied using SEM at a magnification of 1,500×.

Atomic force microscopy

The surface roughness of the samples can be determined with the help of AFM. The AFM model used to analyze the samples is SPA300HV with a scan rate of 1.502 Hz in tapping mode. Here, the surface morphology of mPE, 30 min and 60 min HNO3 treated mPE sample was measured by AFM in contact mode on a 10 ×10μm2 area, and the mean average surface roughness (Ra) and 3D pictographic view is obtained. Each AFM image was analyzed in terms of Ra (Pelagade et al., 2012). The surface roughness is calculated using the software SPIWin.

Tensile testing

The tensile strength was tested using ZWICK Universal Tester (Z010, Germany) at a gage length of 15 mm and a speed of 10 mm-1 for the untreated mPE, 30 min and 60 min HNO3 treated mPE specimens at a load cell capacity of 100 N with a sample thickness of 1 mm. The reported tensile moduli is represented as average results of five tests.

Blood coagulation assays

Prothrombin time (PT)

Prothrombin time is a valuable indicator to find the prohibition of extrinsic pathway. Platelet poor plasma (PPP) (100 µL at 37 °C) was applied on untreated and treated polymer surface with NaCl-thromboplastin (Factor III, 100 mL, Sigma) which contains Ca2+ ions. The time consumed for the formation fibrin clot was assessed with the help of a stopwatch and a steel hook (n = 3) (Amarnath, Srinivas & Ramamurthi, 2006).

Activated partial thromboplastin time (APTT)

APTT is utilized for studying the propensity of the blood to coagulate via intrinsic pathway and to determine the effect of biomaterial on delaying the process. Platelet poor plasma (100 µL at 37 °C) is incubated in prior with substrates at 37 °C and followed by its activation by adding rabbit brain cephalin (100 µL 37 °C). Later, the samples were incubated at 37 °C for 5 min and followed by incubation with calcium chloride (0.025 M). Inclusion of CaCl2 triggers the clotting process. The time taken from the inclusion of CaCl2 up to clot formation is recorded as the activated partial thromboplastin time (APTT) (n = 3) (Amarnath, Srinivas & Ramamurthi, 2006).

Hemolysis assay

The HNO3 treated (30 min and 60 min) and untreated samples were equilibrated with physiologic saline (0.9% w/v; 37 °C, 30 min) followed by its incubation with 3 mL aliquots of citrated blood diluted with saline (4:5 ratios by volume). This mixture of blood and distilled water was prepared at a ratio of 4:5 by volume to result in comprehensive hemolysis which was used as the positive control. Physiological saline solution was utilized as negative control which produces no coloration. The samples were subjected to incubation in their respective mixtures (60 min, 37 °C). These mixtures were later centrifuged and their absorbance of clear supernatant was determined at 542 nm. The absorbance of positive control was normalized to 100% and the absorbance of both the samples was ascertained as a percentage of hemolysis whilst comparing it with positive control (n = 3) (Amarnath, Srinivas & Ramamurthi, 2006).

Platelet adhesion assay

The mPE samples were subjected to HNO3 exposure for 30 min and 60 min, later incubated along with physiological saline (0.9% w/v; 37 °C, 30 min). This is kept on the rotary shaker for an hour to wash of the acid residues on the surface of the polymer. This is followed by immersing of untreated mPE, 30 min and 60 min HNO3 treated mPE samples in 1 mL fresh PRP and the incubation was maintained at 37 °C for an hour. PRP was poured off and the membranes were rinsed in physiologic saline and dried. Ultimately, the samples were viewed using the microscope (n = 3). The polymer surface was photographed and platelet count was determined on a region with a 40× magnification (Amarnath, Srinivas & Ramamurthi, 2006).

Statistical analyses

All experiments were conducted thrice independently. One-way ANOVA was done to determine statistical significance. The results obtained from all experiments are expressed as mean ± SD. In case of qualitative experiments, a representative of three images is shown.

Results

The mean contact angle of the control was found to be 86.06°. This was found to be far greater in comparison to the acid treated samples. The mean contact angles of 30 min and 60 min HNO3 treated samples are 72.03°and 69.73°, respectively, were significantly lower with respect to untreated surface as shown in (Table 1).

Table 1 Contact angle measurement of the mPE before and after HNO3 treatment.

S. No	Sample	Average contact angle in degrees ∗	
1	Untreated mPE	86.06 ± 1.15	
2	mPE treated with HNO3 (30 min)	72.03 ± 2.05	
3	mPE treated with HNO3 (60 min)	69.73 ± 1.41	
Notes.

Values shown are mean ± SD.

∗ Mean differences are significant compared with control (P < 0.05).

FTIR was performed for the determination of chemical composition of untreated and treated samples as shown in Fig. 2. No changes were observed in the functional groups of the untreated mPE, 30 min and 60 min HNO3 treated mPE however there is a slight decrease in the intensity of transmittance is observed. There were alike peaks observed at wavelengths 2,850 cm−1 and 2,930 cm−1 belonging to the alkane group (C–H stretch). The peaks were also found at 3,313 cm−1 (O–H stretching) corresponding to hydroxyl group, 1,647 cm−1 (C = C bending), 1,470 cm−1 (C–H bending) and at 725 cm−1 (C–H rocking), belonging to the alkane family but differ in their structures. A peak was also observed at 1,020 cm−1 which belongs to the C–O stretching.

Figure 2 A representative FTIR spectra of untreated and HNO3 treated mPE.

Figure 3 Different three-dimensional representations using 3D Hirox digital microscopy.

(A) Untreated mPE (B) Untreated mPE with profiling (C) 30 min HNO3 treated mPE (D) 30 min HNO3 treated mPE with profiling (E) 60 min HNO3 treated mPE (F) 60 min HNO3 treated mPE with profiling.

The morphological analysis of the samples was done using the 3D Hirox Microscopy and SEM whereas the nanotopographic analysis of the sample was performed with the help of AFM. The images obtained are shown in Figs. 3A–3F. Figure 3A represents the 3D image of the control and Fig. 3B shows its profiling image. Likewise, Figs. 3C and 3D depict the 3D image and its profiling image of the thirty minutes HNO3 treated sample, respectively. Similarly, Figs. 3E and 3F elucidates the 3D image and its profiling image of one hour HNO3 treated mPE sample. The graph plotted with the values obtained from the 3D-Hirox digital microscope is represented in Fig. 4. Here, each point in the profile line is measured and computed. These points represent the height of the pits in mPE surface. Thus, in Fig. 4, the height of the pits or pores is plotted against the area of the profile line. From this graph, it is palpable that there are fewer pores of pits in case of the control mPE. It is found that the number of pores, increased in the 30 min and found to be highest in 1 h acid treated mPE sample. The 1 h HNO3 treated sample has the maximum number of pores with greater depth of fissures and holes which was ascertained using the 3D profiling. Figure 4 shows the depth of the pores formed due to the etching effect of the HNO3 and it is evident that the control has the least pore depth, followed by 30 min HNO3 treated sample and finally the one hour HNO3 treated sample. Hence, the duration of acid treatment has an impact on the surface porosity by affecting the pore diameter or area. From Fig. 4, it is visible that 42.5 µm was the highest height of the pore in the case of 60 min HNO3 treated sample whereas the highest pore for the 30 min HNO3 treated mPE was 30 µm and 17.5 µm for untreated mPE. This shows the numerical values, data on relative changes for clearly differentiating the etching effect of HNO3 on mPE.

Figure 4 The representative height of the pores of different samples measured using 3D-profiling of 3D Hirox digital microscopy.

SEM imaging is another surface characterization method of the samples at the micro level (Zhao et al., 2011). Topography of the mPE was investigated as shown in Figs. 5A–5C. It was observed that the surface of mPE sample has very less or negligible pits under a 1,500× magnification. However, on observing the SEM image of 30 min treated sample, it was found that the surface of the treated samples has been etched by the acid exposure. A few number of pit formation was also observed. However, the size and the number of the pits seems to increase in case of the 60 min acid treated sample.

Figure 5 Representative SEM micrographs of untreated and HNO3 treated mPE.

(A) Untreated mPE (B) 30 min HNO3 treated mPE (C) 60 min HNO3 treated mPE.

The AFM images are represented in Figs. 6A–6C. Figures 6A–6C are the AFM image of the untreated, 30 min and 60 min HNO3 treated mPE sample, respectively. From the results obtained, it was found that the mean value of Ra of untreated mPE film, 30 min and 60 min HNO3 treated mPE surface are 2.069 nm, 4.233 nm and 5.127 nm, respectively. The nanotopographic analysis of the samples was performed using AFM. Fig. 6A illustrates the 3D surface topography of the sample mPE. Here, it is observed that the surface of the sample is even with fewer hills and valley structures in the untreated mPE sample. On the other hand, Fig. 6B which is the 30 min HNO3 treated sample, has more nano-roughness compared to the control but less roughness than the 60 min HNO3 treated sample. This shows that the 60 min HNO3 treated sample has the most hill and valley structures on the surface of the sample.

Figure 6 Representative AFM images of untreated and HNO3 treated mPE.

(A) Untreated mPE (B) 30 min HNO3 treated mPE (C) 60 min HNO3 treated mPE.

The average tensile testing result of mPE before and after nitric acid treatment is represented in Figs. 7A–7C. From the tensile stress–strain curve it is palpable that the elastic modulus of 30 min HNO3 treated mPE is 33.01 MPa (Fig. 7B) and 60 min HNO3 treated mPE is 34.75 MPa (Fig. 7C) which are slightly greater than the elastic modulus of untreated mPE 31.32 MPa (Fig. 7A). The elastic modulus, maximum force, elongation at maximum force and work up to maximum force is given in Table 2.

Figure 7 Representative tensile testing result of untreated and HNO3 treated mPE.

(A) Untreated mPE (B) 30 min HNO3 treated mPE (C) 60 min HNO3 treated mPE.

Table 2 Tensile testing result of untreated, 30 min and 60 min HNO3 treated mPE.

S. no	Sample	E-modulus MPa	Fmax. N	E-FMax.%	W up to Fmax.Nmm	
1	Untreated mPE	31.32	119.09	2510.82	30680.25	
2	mPE treated with HNO3 (30 min)	33.01	120.63	2510.40	31114.00	
3	mPE treated with HNO3 (60 min)	34.75	121.85	2510.39	32513.68	

Prothrombin time and activated partial thromboplastin time tests were done on the three samples, namely, untreated and 30 min and 60 min HNO3 treated. Their results of PT and APTT were summarized in Figs. 8A and 8B, respectively. Both PT and APTT demonstrated an increase in their value for acid treated samples compared to the control. Mean PT of untreated sample was observed to be 19.23 s, whereas 30 and 60 min HNO3 exposed samples shown 19.86 s and 21.4 s, respectively. Likewise, mean APTT was found to be 105.66 s, 113 s and 136.33 s for untreated, 30 min and 60 min acid treated mPE, respectively. Statistical analysis of the untreated sample with the treated ones using one-way ANOVA insinuates significant differences (P < 0.05) between them for both PT and APTT times after 60 min exposure.

Figure 8 Comparison of prothrombin time (PT), activated partial thromboplastin time (APPT) and absorbance of untreated and HNO3 treated mPE.

(A) The PT of control, 30 min and 60 min HNO3 treated mPE (n = 3) (B) The APPT of control , 30 min and 60 min HNO3 treated mPE (n = 3) (C) The absorbance of control, 30 min and 60 min HNO3 treated mPE (n = 3). Values shown are mean ± SD and ∗ indicating differences in the mean are significant (p < 0.05).

Figure 9 Platelet adhesion assay of untreated and HNO3 treated mPE.

(A) Comparison of the number of platelets adhered untreated, 30 min and 60 min HNO3 treated mPE. Values are expressed as mean ± SD and ∗ indicates difference in the means are significant with P < 0.05 (B) Number of platelets adhered on untreated mPE (C) Number of platelets adhered on 30 min HNO3 treated mPE (D) Number of platelets adhered on 60 min HNO3 treated mPE.

Besides that, hemolysis is an important screening test, which provides quantification of small levels of plasma hemoglobin that may not be assessed under in vivo conditions (Schopka et al., 2010). The hemolysis test was conducted on treating samples and untreated sample for investigating the effect of polymer surface on red blood cells (RBC). Mean absorbance seemed to decrease in the case of treated samples (0.02 and 0.007 for 30 min and 60 min HNO3-treated samples) compared with the untreated (0.05) mPE, indicating lesser damage incurred and interaction between the treated samples and RBC (Fig. 8C). This is because the absorbance is directly proportional to the hemolytic index (HI) of the RBC. Statistical analysis of the untreated as well as acid treated samples (absorbance at 542 nm) using one-way ANOVA ascertained significant differences (P < 0.05) between them after 30 min and 60 min treatment. From the results obtained, it is obvious that the 60 min HNO3 treated mPE is the least hemolytic compared to other samples. Moreover, it was also found that an absorbance value of 60 min HNO3 treated mPE to be in similar trend compared to the one hour HCl treated mPE (Jaganathan et al., 2014b).

Besides HI, the adhesion of platelets on a blood contacting device’s surface could result in coagulation and thrombus formation. Hence, the platelet adhesion test has to be performed to analyze the blood compatibility of blood contacting device (Wenzhong et al., 2008). The number of platelets adhered to a surface of treated polymers was found to be reduced to a great extent compared to the number of platelets which was found in the untreated sample as shown in Fig. 9A. A maximum of 22 platelets was observed on the surface of the untreated samples (Fig. 9B), meanwhile the number of platelets decreased to a maximum of 18 platelets (Fig. 9C) and 15 platelets (Fig. 9D) on 30 min and 60 min HNO3 treated samples, respectively. Statistical analysis of the untreated sample with the treated one (number of platelets adhered) with one-way ANOVA shown significant differences (P < 0.05) between them after 30 min and 60 min treatment.

Discussion

Blood clotting occurs when blood comes in contact with a foreign surface such as implants following platelet activation. This can be catastrophic in clinical settings, especially in case of various biomedical applications like grafts, catheters, hemodialysis, bypass/extracorporeal membrane oxygenation, and ventricular assist devices (Qi, Maitz & Huang, 2013). In order to circumvent this issue, the hemocompatibility of the blood contacting devices has to be improved and HNO3 surface treated mPE holds great potential. For ascertaining the topographical modification caused by HNO3 on the mPE sample, characterization tests was performed using 3D Hirox, SEM, AFM, contact angle and FTIR. On the other hand, the blood compatibility of the sample was studied by conducting different blood coagulation assays like hemolysis assay, PT, APTT, and platelet adhesion.

The decrease in contact angle indicates the improved wettability and hydrophilicity of the mPE polymer. It is a well known fact that the surface energy is a vital parameter determining polymer’s adhesion, material wettability and even biocompatibility (Kwok, Wang & Chu, 2005). Thus, the assessment of contact angles is contemplated as one of the most convenient method for the determination of surface free energy of solid samples. This technique depends on the interactions between the solid sample of interest as well as liquids with well determined surface tensions. Our result is in good agreement with our previous published results of HCl exposed mPE (Jaganathan et al., 2014b). Thus, the improved surface roughness is reflected in the decrease in contact angle with HNO3 treatment time. Furthermore, in a recent work, Gomathi et al. (2012) had performed surface modification of polypropylene by nitrogen containing plasma improved the polymer’s wettability by decreasing the water contact angle and resulted in enhanced biocompatibility and blood compatibility further corroborates our results. According to the Wenzel model, the improvement in the surface roughness of mPE contributes to the reduction in the water contact angle of mPE (Chau et al., 2009). Thus, it indirectly shows that the surface roughness of the mPE sample are improved by the HNO3 treatment, thereby decreasing the contact angle. Ultimately, the hydrophilicity and mPE hemocompatibility is improved where it can serve as a putative blood contacting device (Zhao et al., 2011). Similarly, the improved surface roughness is also palpable from the results of AFM, SEM, Hirox 3D microscopy result. Hence, the decrease in the contact angle and increase in surface roughness are in consensus from obtained result. This is analogous to the studies that have ascertained that the increase in wettability is instigated by the increase in surface roughness and associated decrease in the contact angle (Mirabedini et al., 2004; Mirzadeh & Dadsetan, 2003; Rochotzki et al., 1994).

There is no alteration in the functional groups of the treated and untreated samples which is done by Attenuated Total Reflectance Fourier Transform Infrared Spectroscopy (ATR-FTIR) studies. However, the variation in the intensity of the peaks depicts that there is some morphological changes occurred. The surface changes induced in the metalocene polyethylene (mPE) surface is instigated by the etching effect of nitric acid (HNO3). The HNO3 exposure produces pits and holes in the surface of mPE. Owing to the formation of pits and pores in mPE, the surface roughness increases. The improved surface roughness of mPE is reflected in increased intensity of absorbance in FTIR but the peaks remain almost the same, indicating no chemical changes were seen in the surface of mPE. Bergström (2008) demonstrated that absorption heavily depends upon the surface properties of the material. Since most real life surfaces are not perfectly flat and possess certain degrees of texture and roughness to them, this will influence their optical behaviour. Pits and valleys in a material may, for example, “trap” some of the light and thereby increase the intensity of absorption. However, when the absorption increases, the transmittance decreases (Sorrell, 2006). This explains the reason for the increased absorption leading to decreased intensity of transmittance in this study, as the surface roughness improves after HNO3 treatment. Hence, this elucidates that there is no change in the functional groups in mPE surface, even after HNO3 treatment of mPE which is similar to our HCl exposed mPE (Jaganathan et al., 2014b). Thus, HNO3 treated mPE sample would have enhanced blood compatibility without affecting the chemical structure of mPE since the surface roughness of the mPE is increased by the HNO3 treatment rather than modifying the chemical structure of mPE. The percentage of weight loss study was also performed, but the change in the weight of the sample after HNO3 treatment was not significant which ascertains there is no strong oxidation have occurred to increase the weight of the HNO3 treated mPE samples (result not shown). This is in accordance with the FTIR result which didn’t show appreciable changes in surface functional groups. Thus, it can be elucidated as the improved surface roughness resulted in better hydrophilicity and hemocompatibility of HNO3 treated mPE rather than the change induced in the sample surface functional group by HNO3 treatment.

The 3D Hirox Microscopy images can be interpreted as HNO3 etches the surface of mPE, and one hour acid treatment must have etched the mPE surface more than the 30 min treated sample, thereby resulting in mPE with more pits and pores with higher depth compared to the control. These observations can be compared to a later work of (Vital et al., 2015) where the amplitude of depressions formed in the surface of polymer increased after tetrahydrofuran (THF) and acetic acid surface treatment whereas thickness of the polymer film remained unchanged. Hence, the etching effect of the acid has a favorable impact on the final surface wettability of the polymer, thereby making it more hydrophilic similar to the other surface treatments like plasma treatment to make it blood compatible for various blood contacting device applications (Yue et al., 2015).

Similarly, a larger surface disorientation and improved surface roughness were noticed in 60 min treated sample using SEM images. The SEM images of a recent study show the morphology of isotactic polypropylene (IPP) surfaces of Argon plasma treatment, showing amorphous region is etched on the surface of IPP and the etching depth was found to be increasing with an increase in the time of plasma treated thereby improving its biocompatibility (Ma et al., 2012). Likewise, SEM images of the treated and untreated samples to clearly show that there are pits formed in the surface of the mPE polymer when it is treated with HNO3 and the pit size is also increased with the increase in the time of acid treatment. Since pits were formed, the morphological characteristic such as roughness was also observed to be increased in case of the HNO3 treated samples. It is obvious from Fig. 5 that the number of pits formed in the sample is in the descending order of 60 min HNO3, 30 min HNO3 and then finally the control.

It was found that the Ra value of the 60 min HNO3 treated sample value is almost twice greater than the control and slightly higher than the 30 min HNO3 treated mPE. This is because of more number of hills and valley nano-topographic structure in the mPE sample resulted due to the etching effect of the HNO3 on the mPE sample. This result is found to be similar to a latest study done by Cesca et al. (2014) where the AFM result obtained after poly-3-caprolactone (PCL) etched using mixed gas SF6/Argon at −5 °C has an improved surface roughness resulting in improved biocompatibility. The roughness values obtained using AFM also evidenced the surface structuring after subjecting the sample to surface modification techniques to produce a rougher surface (Tverdokhlebov et al., 2015; Wanke et al., 2011). Similarly, there were other studies carried out show increase in surface roughness of sample results in improved biocompatibility (Slepicka et al., 2013). Hence, AFM nano imaging further bolsters the concept of nanotopographic surface modification caused by the acid etching effect on mPE analyzed using Hirox microscopy and SEM. This formed nanotopographic surface result in improved wettability and hydrophilicity ascertained by contact angle analysis, thereby improving the blood compatibility of mPE which is the cornerstone for blood contacting devices.

Since nitric acid improves porosity and blood compatibility of mPE, the elastic modulus of mPE was studied to make sure that the nitric acid treatment does not deteriorate the elastic modulus of mPE. There was no significant change was observed and minor improvement in the elastic modulus of HNO3 treated mPE which may have resulted due to increase in the roughness of the surface in mPE. This result is in accordance with a recent study where the impact strength of nitric acid treated polyoxymethylene improved compared to those untreated samples (Zhang et al., 2014). The main advantage of the HNO3 over HCl is the improvement of hemocompatibility without deteriorating the tensile strength of the mPE samples after treatment. The effect of HCl on the tensile strength of mPE is not yet reported, but it was found that the HCl treatment deteriorate the tensile strength of the sisal fiber (Oladele, 2010). Similarly, in another work done by Wang et al. (2008) the tensile strength of basalt fiber declines after the HCl exposure. However, the results of HNO3 exposure improved the tensile strength of different polymers like polyacrylonitrile, carbon fibers, thermoplastic polyimide composite and etc., (Bahl, Mathur & Dhami, 1984; Li, 2009; Nie & Li 2010). In our study HNO3 treatment did not reduce the tensile strength of mPE. Hence, the major advantage of HNO3 treatment over HCl treatment is the deterioration of tensile strength of mPE can be prevented, in addition to the enhancement the blood compatibility properties of mPE sample.

Coagulation system activation is triggered by implanting blood contacting device-protein interaction. The activation of factor XII is the first step in this activation process. Reciprocal as well as auto activation will in turn cause the amplification of activated factor XII, where this will initiate the intrinsic coagulation pathway through activation of factor XI, and finally lead to the production of fibrin. Similarly, the activation of platelets by artificial surfaces occurs due to the contact of platelets with artificial surfaces, in terms of ligand expression (GP IIb/IIIa). Ultimately, these activated platelets either adhere to the surface of blood contacting devices through proteins like fibrinogen or aggregate (Schopka et al., 2010). In order to function as a viable blood contacting devices, the implanted blood contacting device should not elicit any unwanted reactions leading to blood clot. In order to investigate that, the blood coagulation assays were carried out in the mPE treated with HNO3. There was a notable increase observed in the PT and APTT of the HNO3 treated mPE sample compared to the control. Changes in surface morphology of mPE by acid treatment helped in improving the blood compatibility of the polymers (Pandiyaraj et al., 2009). Thus, as discussed earlier, the increased PT and APTT is may be attributed by improved surface roughness by the formation of nanotopographic surface by HNO3 on mPE.

The improved surface roughness induces physicochemical changes which results in a favorable impact on the hemocompatible property of mPE thereby making it more resistant to RBCs damage. The lysis of RBCs generally occur due to the increase in osmotic pressure triggered by the poor material surface which normally results in the rupture and release of the cellular contents including hemoglobin (Zhang et al., 2015). Hemolysis percentage is the representative of RBCs damage. Hence, the improved surface roughness of HNO3 treated mPE decreases the RBCs damage when they come in contact with them. According to ASTMF756-00(2000) standard, both 30 and 60 min HNO3 treated samples are deduced to be non-hemolytic materials since the percentage of damage falls below 2 (Fazley, 2014). This result is in agreement to one of the research elucidating that the surface roughness plays an important role in controlling the thrombogenicity and it was demonstrated that catheters with increased roughness were found to be less thrombogenic than smooth surfaced catheters (Bailly et al., 1999). This means that the surface modification of mPE with HNO3 does not induce any damage in erythrocytes’ membranes that could lead to their lysis. Albeit some literatures indicate that it is not possible to define a universal level of acceptable or unacceptable hemolysis values, a blood-compatible material must inhibit hemolysis (Wenzhong et al., 2008). In this study, this parameter is of extreme importance as the proposed mPE material will be in contact with blood for a prolonged period in the blood circulation system.

Figure 10 Mechanism of improved hemocompatibility by HNO3 treatment.

The increased surface roughness of HNO3 treated mPE produces a better resistance to platelets by minimizing the platelet adhesion. The extent of platelet adhesion as well as its activation is a deciding factor of thrombogenecity of a material, as blood compatible substrates neither attracts nor activates the platelets present in the blood stream. In general the hydrophilic surface is observed to be more efficient in preventing the platelet adhesion. This is because it has the ability to exert steric repulsion to avoid unspecific protein deposition (Gomathi et al., 2010). Moreover, the hydrophilic surface is also found to encourage the adsorption of anti-platelet adhesion proteins like albumin, high molecular-weight kininogen, etc., which further bolsters its shielding capacity against platelet deposition and activation. These observations were evidently replicated in the HNO3 treated mPE surfaces (Gomathi et al., 2010; Lee & Lee, 1998). The obtained results are in consensus with other studies conducted on hydrophilic surface dictating that improved surface roughness led to decreased platelet adhesion (Zingg et al., 1981; Zingg et al., 1982). Similarly, Zhao et al. (2011) demonstrated that when the NiTi alloy surface roughness was increased, platelet activation, adhesion, as well as hemolysis were appreciably reduced. The reduced platelet adhesion in the HNO3 treated sample dictates the improved hemocompatibility of surface modified mPE (Gomathi et al., 2012; Habibzadeh et al., 2014).

Thus, the possible mechanism from the obtained result of this study is the surface roughness of mPE improves by the HNO3 treatment. When surface roughness of mPE increases, it results in decrease in contact angle and increases the hydrophilicity of mPE. Hence, the improved surface roughness minimizes the RBCs damage by reduction of osmotic pressure triggered by the poor material surface and leads to decreased platelet adhesion by exerting steric repulsion to avoid unspecific protein deposition. This is the possible mechanism by which the blood compatibility of HNO3 treated mPE improves when it is compared against the control as represented in Fig. 10. Hence, this modified mPE with more surface roughness, altered wettability, and better blood compatibility may be the vital characteristics that can be utilized for construction of long-term blood contacting devices like catheters, transvenous pacing leads, stents, grafts and etc.

Conclusions

The surface modification induced by HNO3 on mPE and its effect on mPE blood compatibility was assessed. Contact angle analysis depicts a decrease in the contact angle elucidating increase in the wettability of the HNO3 treated samples. There were no prominent qualitative changes in the functional groups were observed by FTIR studies. The 3D Hirox microscopy imaging also confirms the improved surface roughness by formation of more pits and bumps in the acid treated sample than the control. SEM images of treated samples further substantiate that acid treated sample surface possess more pits and pores compared to the control. AFM topographical analysis shows an improved surface roughness in the 30 min and 60 min acid treated sample compared to the control due to the etching effect of the acid. Blood coagulation assays like PT and APTT ascertains a notable delay in the clotting mechanism on the surface of acid treated samples. The result of hemolysis assay shows a minimum damage to red blood cells (RBC). Platelet adhesion assay elucidates that the number of platelets adhered to the surface of acid treated polymer was appreciably less in comparison to the untreated surface. The HNO3 treatment of the mPE induces a surface modification in mPE and improves its porosity without much effect on its tensile strength. Hence, HNO3 treated mPE sample can be exploited for various blood contacting biomaterial applications due to its improved blood compatibility.

Supplemental Information

Supplemental Information 1 Comparison of prothrombin time (PT) of control and HNO3-treated metallocene polyethylene (n = 3)

PT demonstrated an increase in their value for acid treated samples compared to the control. Mean PT of untreated sample was observed to be 19.23 s, whereas 30 and 60 min HNO3 exposed samples shown 19.86 s and 21.4 s, respectively indicating improved hemocompatibility.

Click here for additional data file.

Supplemental Information 2 The activated partial thromboplastin time (APPT) of control and HNO3-treated metallocene polyethylene (n = 3)

Mean APTT was found to be 105.66 s, 113 s and 136.33 s for untreated, 30 min and 60 min acid treated mPE, respectively elucidating improved blood compatibility.

Click here for additional data file.

Supplemental Information 3 The absorbance of control and HNO3-treated metallocene polyethylene (n = 3)

Mean absorbance seemed to decrease in the case of treated samples (0.02 and 0.007 for 30 min and 60 min HNO3-treated samples) compared with the untreated (0.05) mPE, indicating lesser damage incurred and interaction between the treated samples and RBC.

Click here for additional data file.

Supplemental Information 4 Platelet adhesion assay of untreated and HNO3-treated metallocene polyethylene (n = 3)

The number of platelets adhered to a surface of treated polymers was found to be reduced to a great extent compared to the number of platelets which was found in the untreated sample. A maximum of 22 platelets was observed on the surface of the untreated samples, meanwhile the number of platelets decreased to a maximum of 15 platelets on 60 min treated samples.

Click here for additional data file.

Supplemental Information 5 Contact angle measurement of the mPE before and after HNO3 treatment (n = 3)

The mean contact angle of the control was found to be 86.06°. This was found to be far greater in comparison to the acid treated samples. The mean contact angles of 30 and 60 min treated samples are 72.03°and 69.73°, respectively, dictating improved hydrophilicity to improve blood compatibility of mPE.

Click here for additional data file.

The authors would like to thank MSI Technologies (M) Sdn. Bhd and Progene for providing us KH-8700 3D Hirox Microscope for imaging our samples. The authors would also like to thank Zwick International for helping us analyze the tensile data.

Additional Information and Declarations

Competing Interests

Author Contributions

Human Ethics

Data Availability

The authors declare there are no competing interests.

Muthu Vignesh Vellayappan conceived and designed the experiments, performed the experiments, analyzed the data, contributed reagents/materials/analysis tools, wrote the paper, prepared figures and/or tables.

Saravana Kumar Jaganathan conceived and designed the experiments, performed the experiments, analyzed the data, contributed reagents/materials/analysis tools, prepared figures and/or tables, reviewed drafts of the paper.

Ida Idayu Muhamad contributed reagents/materials/analysis tools, reviewed drafts of the paper.

The following information was supplied relating to ethical approvals (i.e., approving body and any reference numbers):

Prior to blood procurement, the written consent form was given to the healthy volunteers. They read the benefits and risks of participation before expressing his/her willingness by signing the form. All protocols of blood procurement and consent procedure were approved by the Pacheri Sri Nallathangal Amman (PSNA) College of Engineering and Technology Ethical Committee of Dindigul with an approved IRB number: H30114.

The following information was supplied regarding data availability:

The raw data is supplied in the Supplemental Information.

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
