# Peer review of "Unravelling the potential of nitric acid as a surface modifier for improving the hemocompatibility of metallocene polyethylene for blood contacting devices"

_PeerJ, doi:10.7717/peerj.1388_

## Round 0.1 · original submission · Minor Revisions

· Academic Editor

Minor Revisions

Authors must strongly consider the reviewers comments in the new manuscript revised version

Reviewer 1 ·

Basic reporting

The subject is of great importance.
Report is well done.

Experimental design

There remains a question, because no chemical changes were seen on the surface of the PE.
However the morphology changed significantly.
Shoud one perform other chemical analysis method of the surfaces like XPS?

Validity of the findings

There seems to be achieved a positive outcome.
However the explanation of that remains somewhat unclear , since no chemical changes of the surfaces has been seen.
Frome where does the surface roughness increment be a consequence?

·

Basic reporting

The primary aim of this submitted article by M.V. Vellayappan, S.K. Jaganathan and I.I. Muhamad is to evaluate the effect of nitric acid treatment on the metallocene polyethylene (mPE) towards improving its hemocompatibility. Conducting a similar set of experiments, the authors recently reported that hydrochloric acid treatment on the mPE can enhance blood compatibility of mPE compared to the untreated mPE sample (Jaganathan et al, 2014). Based on this background work, in the current research, the authors found that surface treatment induced by nitric acid can also improve the hemocompatibility of mPE and provided relevant experimental data.

Experimental design

The submitted manuscript describes original research within the Scope of the journal.

Validity of the findings

The authors carefully investigated the research questions within the Scope of the present topic. Conclusions were connected to the research questions investigated and based on the current experimental findings.

Comments for the author

The submitted article meets the PeerJ criteria and hence can be accepted.

Reviewer 3 ·

Basic reporting

No Comments

Experimental design

some tests were not include all the control and experimental samples.

Validity of the findings

No Comments

Comments for the author

The manuscript developed a simple and useful method to improve the hemocompatibility of metallocene polyethylene. However, before publication, The followings should be done.
1. all the tests and evaluations should be include all the samples of untreated ,30min and 60 min treated samples. Such as FTIR, AFM, tensile testing, platelet adhesion were not included in the text.
2. Materials and Methods
In Sample Preparation and Acid Treatment section;
WHY authous mentioned that “ The samples subjected for a lesser duration…….” Is it relative with the text?
3. what is advantage for HNO3 treatment over HCl treatment?
4. It is the best to give some mechanism suggestion.

---

## Round 0.2 · accepted · Accept

· Academic Editor

Accept

The authors have made the minor changes required by the referees in their revised version